# Dimensional Analysis of Double-Track Microstructures in a Lithium Niobate Crystal Induced by Ultrashort Laser Pulses

Yulia Gulina [1], Jiaqi Zhu [1,*], Alexey Gorevoy [1], Mikhail Kosobokov [2], Anton Turygin [2], Boris Lisjikh [2], Andrey Akhmatkhanov [2], Vladimir Shur [2] and Sergey Kudryashov [1,2]

1   Lebedev Physical Institute, 119991 Moscow, Russia; gulinays@lebedev.ru (Y.G.); a.gorevoy@lebedev.ru (A.G.); kudryashovsi@lebedev.ru (S.K.)
2   Institute of Natural Sciences and Mathematics, Ural Federal University, 620000 Ekaterinburg, Russia; mihail.kosobokov@urfu.ru (M.K.); anton.turygin@urfu.ru (A.T.); boris.lisikh@urfu.ru (B.L.); andrey.akhmatkhanov@urfu.ru (A.A.); vladimir.shur@urfu.ru (V.S.)
*   Correspondence: ch.czyaci@lebedev.ru

**Abstract:** Double-track microstructures were induced in the bulk of a z-cut lithium niobate crystal by 1030 nm 240 fs ultrashort laser pulses with a repetition rate of 100 kHz at variable pulse energies exceeding the critical Kerr self-focusing power. The microstructure topography was characterized by atomic force microscopy in piezoelectric response mode. The spatial positions of laser-induced modification regions inside lithium niobate in the case of laser beam propagation along the crystal optical axis can be directly predicted by simple analytical expressions under the paraxial approximation. A dimensional analysis of the morphology of the double-track structures revealed that both their length and width exhibit a monotonous increase with the pulse energy. The presented results have important implications for direct laser writing technology in crystalline dielectric birefringent materials, paving the way to control the high spatial resolution by means of effective energy deposition in modified regions.

**Keywords:** lithium niobate; uniaxial birefringent crystal; femtosecond laser pulses; direct laser writing; Gaussian beam focusing; nonlinear optical interaction





## 1. Introduction

Lithium niobate ($LiNbO_3$) is a uniaxial birefringent material known for its remarkable electro-optic, acousto-optic, and nonlinear optical properties [1]. Femtosecond laser writing in lithium niobate has enabled the development of various techniques for modifying and fabricating structures within this uniaxial birefringent material. For example, optical waveguides created in bulk of $LiNbO_3$ can be used for integrated optics [2], photonic circuits [3], and quantum information processing [4]. Complex photonic structures [5] can be created within $LiNbO_3$, such as Bragg gratings [6] or resonators [7]. The control of ferroelectric domain structures within $LiNbO_3$ can be achieved through laser-induced electric fields, allowing the writing or erasing of domain structures in the crystal [8]. The precise fabrication of micro- and nano-scale structures in $LiNbO_3$ has many applications in micro-optics, micro-electromechanical systems (MEMS) [9], and plasmonics [10]. It can also be used to manufacture sensors [11]. These key techniques take advantage of the ultrafast nature and high peak power of femtosecond laser pulses to generate precise and localized modifications within the crystal, showcasing the potential of femtosecond laser writing for precise manipulation and processing of lithium niobate.

Femtosecond lasers are widely used in the field of micro/nano processing due to their unique advantages such as a significantly reduced thermal effect, a processing accuracy that can break through the diffraction limit, and three-dimensional processing inside transparent materials. Ultrashort high energy laser pulses are able to be absorbed in transparent media such as lithium niobate due to non-linear multiphoton absorption. As a

result, it becomes possible to localize the effect inside the volume of the crystal, which opens the way for the creation of intra-volume optical elements based on the nonlinear optical properties of $LiNbO_3$, 2D and 3D nonlinear optical gratings for second harmonic generation in bulk crystals [12], and 1D nonlinear gratings in waveguides [13]. When high-energy femtosecond laser pulses pass through transparent media (such as gases, liquids, or solids), the laser beam forms tightly packed, elongated optical structures, creating one or multiple tracks. The primary mechanism behind track formation is the self-focusing of the light beam and the nonlinear refractive index changes in the medium. The competition between the self-focusing Kerr effect and diffraction leads to the laser beam focusing in an elongated region, triggering other nonlinear optical processes, such as multiphoton absorption and plasma generation [14]. This, in turn, has a direct impact on the energy deposition area and material modification region. Therefore, by adjusting the parameters of the focused laser pulse (e.g., pulse energy, pulse width, wavelength, numerical aperture, pulse duration, and polarization state [14–18]) and using specific optical components (e.g., lenses, gratings, and wave plates [19]), the control and optimization of modification regions can be achieved.

A linearly polarized focused Gaussian laser beam that propagates from a homogeneous medium into a uniaxial birefringent crystal either along or perpendicular to the crystal's optical axis typically has two or three focal points [20]. When the focused light intensity of the laser pulse approaches the modification threshold and the peak power does not exceed the critical Kerr self-focusing power, a set of analytical expressions for the paraxial electric fields can be used to theoretically estimate the area of laser modification in $LiNbO_3$, as well as the positions of the multiple focal points caused by birefringence [20,21]. The double-focusing phenomenon, when the beam propagation direction is along the optical axis, results from the axial separation of both the amplitude and Gouy phase shift of ordinary (O) and extraordinary (E) waves. The birefringence of the crystal plays an important role in determining the morphology of the structures, so it can lead to the formation of highly anisotropic features. The formation of multiple foci or regions of high intensity was observed, as well as a long track that extends along the propagation axis of the pulse [22]. A detailed analysis of axial focus splitting induced by birefringence in $LiNbO_3$, the dependence of focus splitting characteristics on the polarization and wavelength of the incident beam, and the focusing depth and orientation of the crystal have been provided [23]. However, despite a lot of research in the field of direct laser writing in lithium niobate in the pre-filamentary regime, when peak power does not exceed the self-focusing threshold, the issue of assessing the effect of experimental parameters on the process of structural modification by high-intensity femtosecond laser pulses, when nonlinear optical interaction effects have a significant impact on energy deposition regions, is still relevant.

In this study, a lithium niobate z-cut plate was used to study the non-linear propagation of femtosecond laser pulses along the optical axis, and two sets of track microstructures were induced inside the crystal. The dimensional parameters of the modified area were analyzed and compared with the spatial distribution of the Gaussian beams near the focal points calculated using the analytical expressions.

## 2. Materials and Methods

Spatial structures were generated in the bulk of a congruent lithium niobate (CLN) crystalline z-cut plate using the method of direct laser writing. For laser-induced bulk micropatterning of the CLN crystal, a 3D-micro/nanostructuring laser workstation based on a femtosecond Yb-doped solid-state regenerative femtosecond amplification ultrafast laser system (TETA-10, Avesta Project, Moscow, Russia), with the fundamental wavelength $\lambda$ = 1030 nm (TEM00), full-width at half-maximum (FWHM) pulse duration $\tau \approx 240$ fs, variable pulse energy $E$ = 4–9 µJ, and repetition rate $f$ = 100 kHz, was employed (Figure 1).

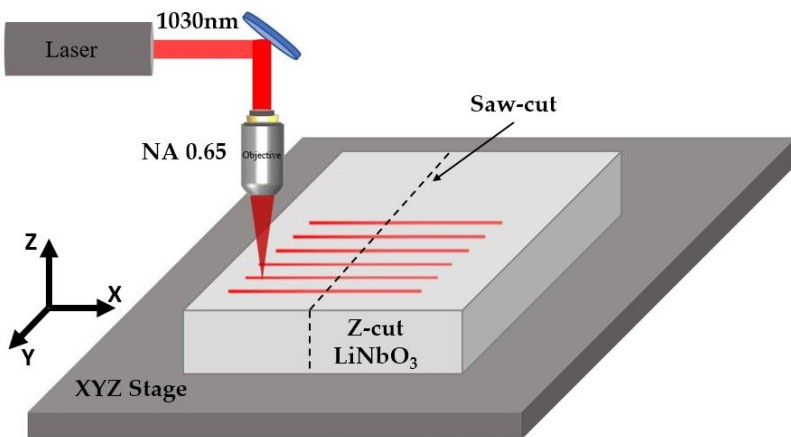

**Figure 1.** Schematic diagram of the laser writing operation in a lithium niobate crystal.

The laser pulses were focused by a 0.65 numerical aperture (NA) micro-objective lens into the $1/e^2$ intensity radius $w_0 \approx 0.82$ μm at a depth of ~500 μm inside the 1 mm thick z–cut lithium niobate plate; thus, the radiation was directed along the optical axis coinciding with the Z-direction, as shown in Figure 1. The sample was mounted on a PC-driven high-precision three-dimensional (XYZ) motorized micro-positioning translation stage (Prior Scientific, Fulbourn, UK) and scanned at a translation speed of 2 mm/s along the X-crystal axis, enabling inscription at different delivered effective energies $E$ = 2–6 μJ taking into account the output efficiency 67% (peak power $P \approx 10$–24 MW) in the non-linear focusing regime ($P > P_{crit} = 0.9 \pm 0.1$ MW, where $P_{crit}$ is the critical Kerr self-focusing power for z-cut CLN at 1030 nm [23]).

In order to reveal the fine topography of the laser-modified CLN regions, the inscribed linear horizontal arrays of vertical nanopatterns in the bulk CLN were saw-cut across the scan lines by an automated precision dicing saw DAD 3220 (DISCO, Tokyo, Japan) using a Disco diamond blade disk Z09-SD3000-Y1-90 55 × 0.1 A2X40-L (DISCO). The cuts were consequently grinded by $Al_2O_3$ powder (grain sizes: 30, 9, and 3 μm) and polished by ≈25 nm colloidal $SiO_2$ nanoparticles on the polishing machine PM5 (Logitech, Windsor, UK) until optical surface quality was obtained. Then, the uncovered topography was characterized by an atomic force microscope (AFM) NTEGRA Aura (NT-MDT, Moscow, Russia) in piezoelectric response mode using Pt-coated NSC 18 probes (MikroMash, Moscow, Russia, tip size 30 nm, first resonance frequency 400–500 kHz and stiffness coefficient 2.8 N/m) at the 10-V, 20-kHz probing AC voltage.

## 3. Results and Discussion

### 3.1. Formation of Double-Track Structures

Previous studies have pointed out that the analysis of the two-dimensional light intensity distribution near the geometrical focal plane in the uniaxial crystal can be divided into two cases, where the laser beam propagates along or perpendicular to the optical axis [20–22,24]. We refer to the analytical expressions derived by P. Karpinski et al. [21]. The propagation model is based on the paraxial wave equation for a linear medium, and it allows direct evaluation of the spatial distribution of focal points inside the crystal. We apply this model for our case of a z-cut lithium niobate plate and the laser beam propagation direction along the optical axis. In this case, the incident Gaussian beam is split into two beams after entering the crystal (Figure 2), resulting in the formation of two waists with different Rayleigh lengths. Assuming that the incident beam is polarized along the $x$-axis, the Cartesian components $E_x$ and $E_y$ of the transverse electric field are expressed as a superposition of two beams with Gaussian envelopes $G_1$ and $G_2$, as follows:

$$E_x = \frac{1}{2}\left[(G_1 + G_2) - \left\{(1 + r^{-2}\omega_0^2\xi_1)G_1 - (1 + r^{-2}\omega_0^2\xi_2)G_2\right\}\cos 2\varphi\right]e^{i(kn_0 z - \omega t)},$$
$$E_y = -\frac{1}{2}\left\{(1 + r^{-2}\omega_0^2\xi_1)G_1 - (1 + r^{-2}\omega_0^2\xi_2)G_2\right\}\sin 2\varphi\, e^{i(kn_0 z - \omega t)}, \tag{1}$$

where $G_1 = (E_0/\xi_1)\exp\{-r^2/(\omega_0^2\xi_1)\}$, $G_2 = (E_0/\xi_2)\exp\{-r^2/(\omega_0^2\xi_2)\}$, $\xi_1 = 1 + i\lambda(z - c_1)/(\pi n_0\omega_0^2)$, $\xi_2 = 1 + i\lambda n_0(z - c_2)/(\pi n_e^2\omega_0^2)$, $r^2 = x^2 + y^2$ is the radial coordinate, $\omega_0$ is the waist radius of the incident beam, $k$ is the wavenumber in vacuum, $\varphi = \arctan(y/x)$ is the polar angle measured from the $x$-axis, $E_0$ is the peak amplitude of the electric field, $n_0$ and $n_e$ correspond to the ordinary and extraordinary refractive index, respectively, and $c_1$ and $c_2$ are constants that determine the position of the waists for $G_1$ and $G_2$ relative to a reference point with $z = 0$. This point is chosen between the two waists (two focal planes), so that $c_1 = c_2 = a$, as shown in Figure 2, and the distance from the front face of the crystal to this point along the $z$-axis is equal to $d$.

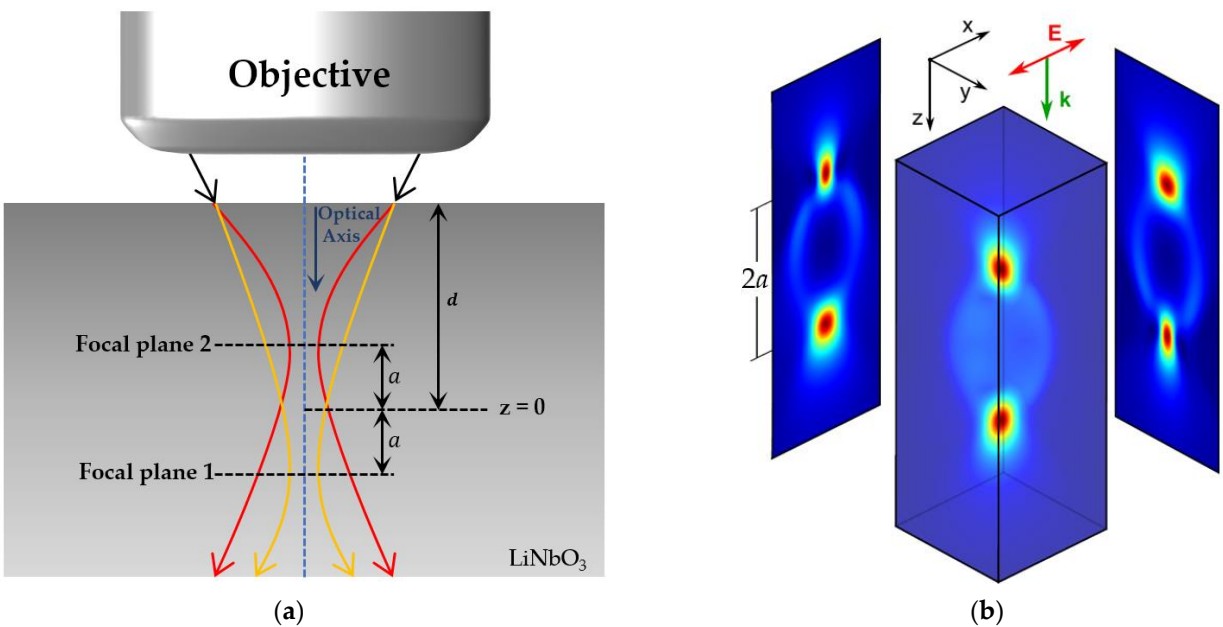

**Figure 2.** Axial focus splitting of a linearly polarized Gaussian beam incident along the optical axis in a lithium niobate crystal: (**a**) diagram and (**b**) simulated intensity distribution of the transverse electric field.

In a LiNbO$_3$ crystal, the ordinary refractive index is greater than the extraordinary refractive index ($n_0 > n_e$). According to the Sellmeier formula [25], $n_0 = 2.234$ and $n_e = 2.158$ at $\lambda = 1030$ nm. The intensity distribution of the transverse electric field calculated according to Equation (1) for $d = 500$ μm is shown in Figure 2b. In this case, the distance from the waists to the reference point is estimated as

$$a = d(n_0^2 - n_e^2)/(n_0^2 + n_e^2) \approx 17.3 \text{ μm}. \tag{2}$$

The Rayleigh lengths of the two Gaussian envelopes can be calculated as

$$Z_{R1} = n_o \cdot Z_R = n_o \cdot \frac{\pi\omega_0^2}{\lambda} = 4.58 \text{ μm}, \tag{3}$$

$$Z_{R2} = \frac{n_e^2}{n_o} \cdot Z_R = \frac{n_e^2}{n_o}\frac{\pi\omega_0^2}{\lambda} = 4.28 \text{ μm},$$

where $Z_{R1}$ corresponds to the beam waist formed furthest from the crystal front face and $Z_{R2}$ is the Rayleigh length of the other beam.

The presence of the two foci and two waists formed in the bulk of the lithium niobate crystal manifests itself as two elongated laser-modified areas that are clearly visible in dark-field optical microscopy (OM) (Figure 3a). Scanning probe microscopy allowed to reveal heterogeneous microtracks of material damage in the foci (Figure 3b). The PFM

(piezoresponce force microscopy) mode demonstrates a change in the PFM amplitude in the foci of the laser radiation. This change may indicate both the presence of domain walls and local degradation of the piezoelectric properties as a result of crystal damage (Figure 3c). A comparison of the topography and PFM images shows the exact localization of the change in PFM amplitude signal in the microtracks. Thus, the image data of the PFM signal amplitude distribution provided a more accurate estimation of the area affected by laser radiation because of the better homogeneity and contrast.

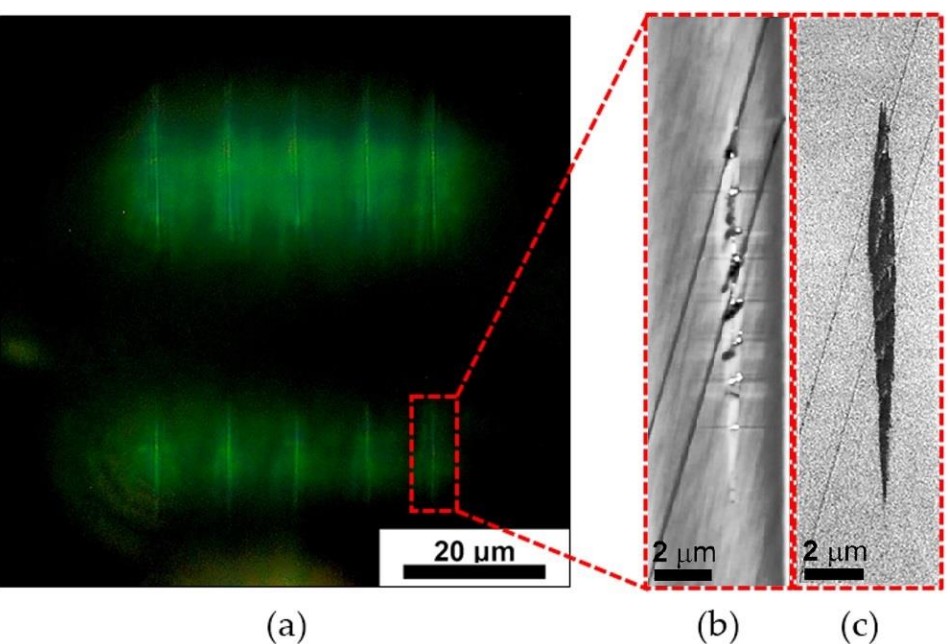

**Figure 3.** (**a**) OM dark field image of the double-track cross-section in lithium niobate under a laser pulse energy of 3.3 µJ, (**b**) AFM topography, and (**c**) PFM amplitude scan cross-sectional images of one of the foci of the laser source.

The results of computer simulations and the obtained PFM image of microstructural changes produced in lithium niobate under a pulse energy of 2.7 µJ are shown in Figure 4. The images clearly show the presence of two distinct laser-modified regions aligned along the writing beam propagation direction at $d \approx 500$ µm focusing depth, as predicted. The top and bottom tracks are induced by two Gaussian beams with effective refractive indices of $n_e^2/n_o$ and $n_o$, respectively. A small mark between the tracks is formed by second harmonic radiation at 515 nm, which was not filtered completely. According to the above theoretical calculations, we measured and marked the distance $2a = d_f$ between the two focal points, as well as the corresponding Rayleigh ranges $2Z_{R1}$ and $2Z_{R2}$ and the beam waist width. It can be seen that the two modified regions have an elongated drop-shaped form due to the nonlinear character of laser radiation–matter interactions. The length of the modified region significantly exceeds the Rayleigh length, and its shape is not symmetric relative to the waist position, which is typically caused by the following processes. During the initial ignition, a thin, straight plasma channel emerges around the focal region (at focal planes 1 and 2). This ionization process is mainly triggered by non-linear absorption (tunnel and multiphoton ionization) [26]. Subsequently, the plasma seems to grow towards the incoming beam in a negative z direction and acquires an elongated shape. Then, the high-energy part of the laser pulse couples with the existing plasma and deposits its energy within the beam caustic, leading to a drop-shaped plasma formation and elongated regions of material modification accordingly.

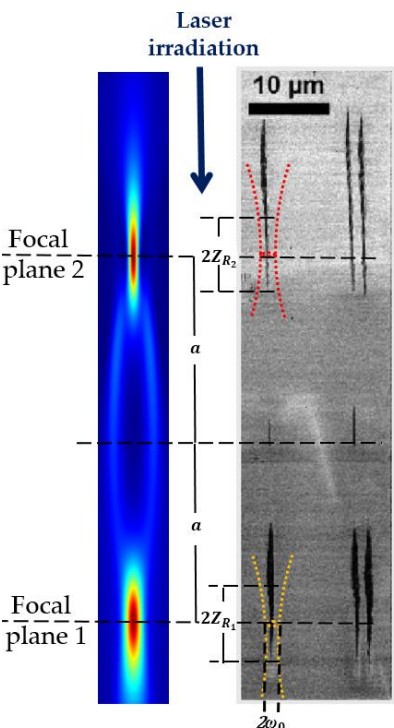

**Figure 4.** PFM amplitude scan image of cross-sectional microstructural changes produced in lithium niobate under a laser pulse energy of 2.7 μJ (**right part**) and computer simulated intensity distribution (**left part**).

*3.2. Dimensional Parameters of Double-Track Structures at Varied Pulse Energies*

The morphology of the modified regions induced in birefringent LiNbO$_3$ crystal under varied pulse energies is shown in Figure 5. All the images demonstrate two sets of elongated drop-shaped tracks incipient from the two split focal planes. We processed the images of tracks for different delivered effective energies of $E$ = 2–6 μJ and calculated the mean values and standard deviations of basic parameters such as the length (along the direction of propagation) and width (perpendicular to this direction) of the tracks.

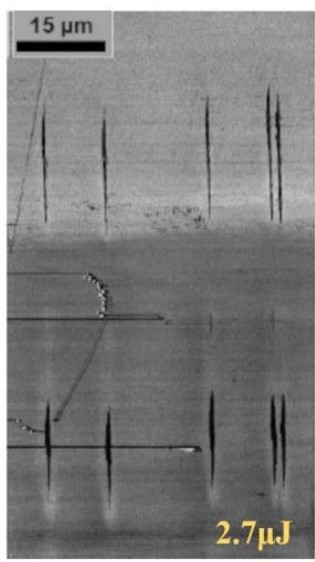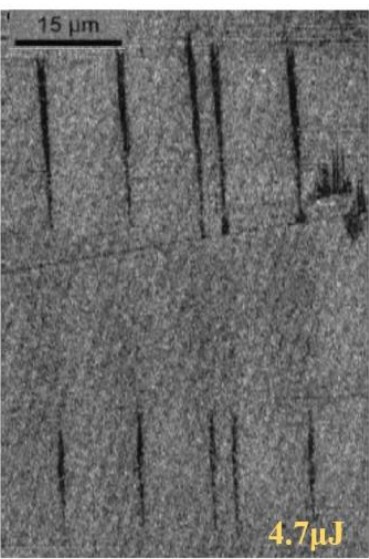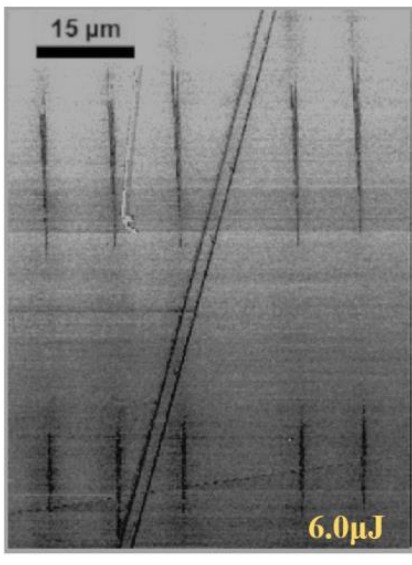

**Figure 5.** PFM amplitude scan images of the internal fine structure in a lithium niobate crystal induced by femtosecond pulses with energies of 2.7 μJ, 4.7 μJ, and 6 μJ.

The measured lengths and widths of the first and second tracks formed near the corresponding focal planes inside the crystal are presented in Figure 6. As can be seen, the length of both tracks increases monotonously with increasing pulse energy, and the second tracks are significantly longer than the first ones. This facilitates the improved control of the length of the filament track by varying the pulse energy. Due to the nonlinear character of the radiation absorption in the bulk of the material, as well as the scattering of radiation on the forming plasma in the second focal area, the intensity in the first focal area reduces, which leads to a decrease in the energy deposition area and a smaller size of the modification region. The tracks width grows slightly with an increase in the pulse energy, and at an energy exceeding 5 μJ, width stabilization is observed. This may be a consequence of the occurrence of dense reflecting plasma in the focal region, which causes energy deposition termination and modification area limitation.

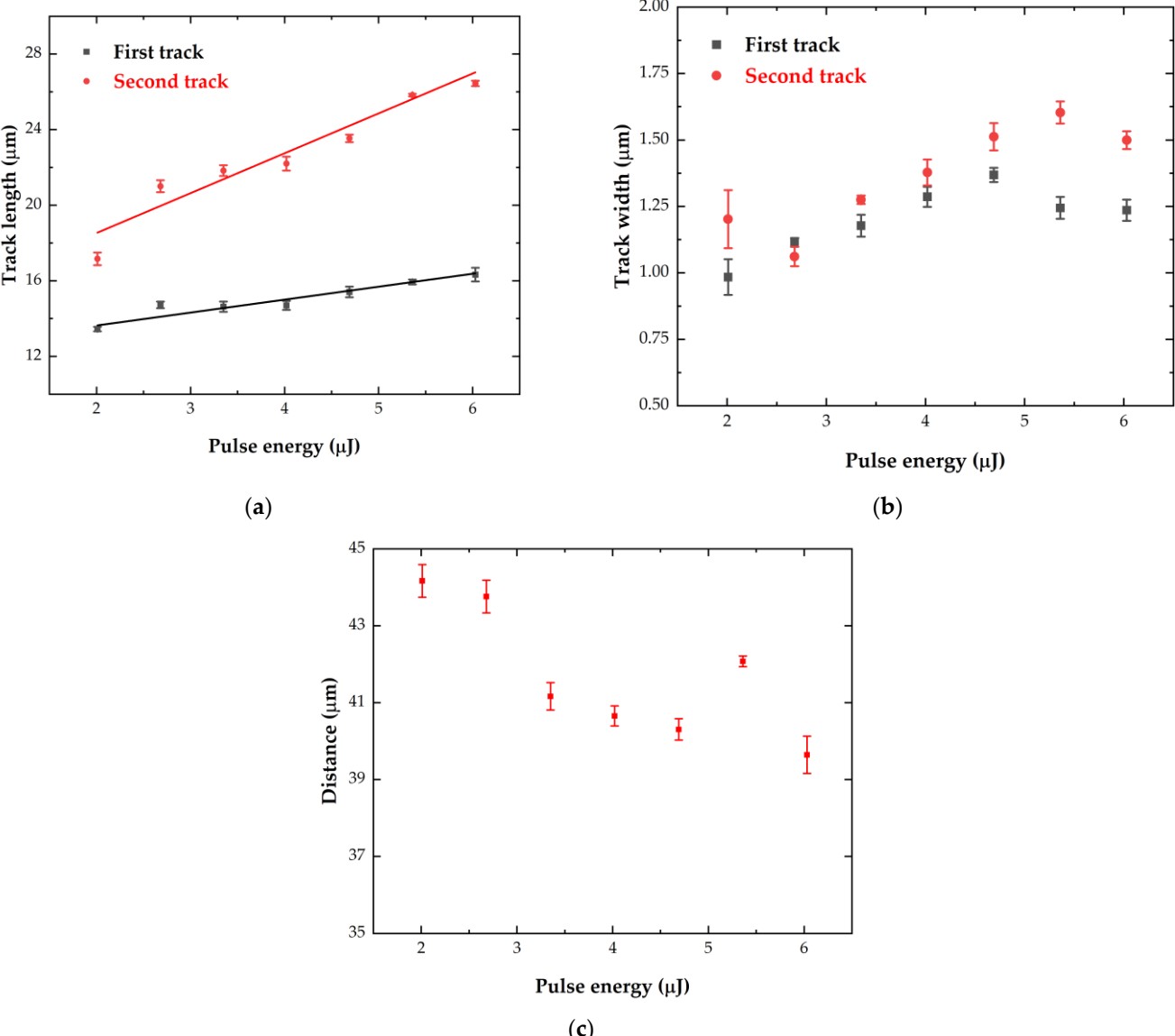

**Figure 6.** Dependence of the track length (**a**), the track width (**b**), and the distance between tracks (**c**) on the pulse energy. A linear fit is shown in (**a**) for reference.

The experimentally obtained dependence of distance between the back points of tracks inside the crystal on the pulse energy is shown in Figure 6c. The separation of the laser-induced modification regions is close to the theoretically predicted value, although the

accurate localization of the waist positions in the images is difficult because of the asymmetry of the modified regions. The measured distance slightly decreases with increasing pulse energy. This is a consequence of the fact that an increase in energy leads to a different elongation of the top and bottom tracks length from the focal plane towards the laser beam propagation.

## 4. Conclusions

The investigation of structural modifications in the bulk of a uniaxial birefringent lithium niobate z-cut plate induced by femtosecond laser pulses in the non-linear focusing (filamentation) regime has shown that beam propagation along the crystal optical axis results in the production of double-track structures, depending on the incident pulse energy. It was demonstrated that that spatial positions of the laser-induced modification regions inside lithium niobate can be directly predicted by simple analytical expressions under a paraxial approximation. A dimensional analysis of the double-track structures revealed that both their length and width exhibit a monotonous increase with the pulse energy. The obtained results are important not only for microprocessing of lithium niobate single crystals but also for high-NA focusing micromachining of any crystalline dielectric birefringent material. Even if the pump beam is guided along the optical axis in these materials, a pair of tracks of similar size at different depths instead of one will be obtained. This fact must be taken into account during the creation of functional 3D gratings in these materials. Therefore, the presented results have important implications for the whole field of direct laser writing in crystalline dielectric birefringent materials, showing that the spatial resolution of the induced structures can be significantly decreased by multi-focusing effects and other distortions of the focal intensity distribution. On the other hand, the obtained double-track formation can potentially be used for effective double-focus microstructuring of crystalline dielectric birefringent materials to increase the processing speed.

**Author Contributions:** Conceptualization, S.K., Y.G., A.A. and V.S.; methodology, S.K. and Y.G.; software, A.G. and A.T.; validation, Y.G., S.K. and M.K.; formal analysis, Y.G., J.Z. and A.G.; investigation, M.K. and B.L.; resources, M.K., V.S. and S.K.; data curation, J.Z. and A.T.; writing—original draft preparation, Y.G., J.Z., A.G. and M.K.; writing—review and editing, Y.G., A.G. and S.K.; visualization, A.T.; supervision, S.K., A.A. and V.S.; project administration, S.K., A.A. and V.S.; funding acquisition, A.A. and V.S. All authors have read and agreed to the published version of the manuscript.

**Funding:** This research was funded by the Ministry of Science Higher Education of the Russian Federation (state task FEUZ-2023-0017).

**Institutional Review Board Statement:** Not applicable.

**Informed Consent Statement:** Not applicable.

**Data Availability Statement:** The data presented in this study are available on request from the corresponding author.

**Acknowledgments:** The equipment of the Ural Center for Shared Use "Modern nanotechnology" Ural Federal University (Reg.№ 2968) which is supported by the Ministry of Science and Higher Education RF (Project № 075-15-2021-677) was used.

**Conflicts of Interest:** The authors declare no conflict of interest.

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
