# Peer review of "Dimensional Analysis of Double-Track Microstructures in a Lithium Niobate Crystal Induced by Ultrashort Laser Pulses"

_photonics, doi:10.3390/photonics10050582_

Round 1

Reviewer 1 Report

The manuscript "Dimensional analysis of double-track microstructures in lith-2 ium niobate crystal induced by ultrashort laser pulses" by Yulia Gulina et al. reports on the effect of experimental parameters on the process of structural modifications by high-intensity femtosecond laser pulses pose in bulk uniaxial birefringent lithium niobate Z-cut crystal. The main finding is the creation of double-track structures depending on the incident pulse energy deposited on the crystal. The authors have also incorporated analytical expressions that agree with the experimental findings. 

Overall, the manuscript is well written, easy to read and follow, the experimental procedure is given in detail, the subject of research is interest for the broad readership of MDPI Photonics and fits the scope as well. I recommend it for publication only after my comments are addressed.

- In Fig. 6 and relevant text, lines 196 -213:  This part is not explained as accurately as the rest of the text. I am a bit confused, is there a difference between bottom and top tracks and first and second focal areas? If yes clarification should be given. If not, I would suggest to keep one of the two. 

-In Fig. 6(a) the length seems to increase linearly with pulse energy in both tracks. Does this linearity has a physical meaning? Would there be a scenario where nonlinear increase would be present?

-In Fig. 6(b) line 202-203 "The tracks width also increases with pulse energy." This not true for all pulse energies. There seems to be a threshold around 5.5μJ for the top track and 5μJ for the bottom where after these values a decrease occurs. The authors have drawn linear fits to the data. However, the spread of these data does not support this linearity for all values. For pulse energies ~>5μJ a deviation from linearity is observed. This should be mentioned in the text. Also, is there a physical meaning or other explanation for this behaviour? Having measurements for greater pulse energies would help. 

- Fig. 6(c). Again, this linear fit seems out of context. I see many other possible fits of these data. Why is this linearity important?

- In all previous cases, does the slope of the linear fits mean something? 

- In the conclusion section: More specific reasoning on why is the obervance of this double-track structure and its implications for the field of direct laser writing should be given. The word "important" in line 223 on its own does not say much. Is it important only for lithium niobate crystals? Could it be important also to other crystals for instance? More justification is needed.  

Reviewer 2 Report

The authors created double-track microstructures in a bulk lithium niobate crystal with a femtosecond laser beam. The microstructure geometry variation behavior, including length, width, and distance, with respect to rising laser pulse energy was revealed and analyzed with an analytical model to account for such a non-linear double-focusing phenomenon under the paraxial approximation. The numerical results qualitatively agreed with the experimental observations. The manuscript is well-organized overall and matches the interested fields of this journal. However, before this manuscript becomes acceptable, the authors are required to address the following comments well.

1, The impact of your study should be strengthened in the abstract and conclusion part.

2, In the manuscript, “z-cut” and “Z-cut” are used together. Please only use one format and make them consistent.

3, In line 90, “NA” is used for the first time in the manuscript. Please add the full name of NA (numerical aperture).

4, Please clarify the meaning of “Z - - side” in line 92.

5, In line 103, “Al2O3” should be replaced by “Al2O3”.

6, In line 124 and 125, please modify the index of the equations. In line 142 the authors mentioned equation (1). However, there is no equation (1) in the manuscript.

7, In Fig. 5(a) and (b), there are two microstructures formed very closely. Are they induced by one laser beam or two laser beams?

8, If the authors use even higher pulse energy, what will happen to the length, width, and distance of such two microstructures?

9, If the authors process the same region twice, what will happen to the length, width, and distance of such two microstructures?

Based on the abovementioned comments, this manuscript is recommended for minor revision, and a revised manuscript is required.

Author Response

Please see bthe attachment.

Reviewer 3 Report

The manuscript by Gulina et al. reports on the morphological characterization of bulk microstructures induced in lithium niobate crystal by a femtosecond pulsed laser. Interestingly, when the high energy pulses are focused inside the crystal, they form a couple of tightly-packed, elongated optical structures, that the authors defined “tracks”. As a result, the authors present a dimensional analysis of the tracks morphology in cross-section. The analysis was carried out by AFM, and reveal that both depth and width of the produced tracks exhibit a monotonous increase in relation to the used laser pulse energy.

Lastly, it was demonstrated that the spatial positions of laser induced modification regions inside the crystal can be directly predicted by analytical expressions under paraxial approximation.

General comments

The manuscript is written in a clear way, using a good English. The work is interesting and presents a solid analytical background. Furthermore, the topic of Femtosecond Laser Micromachining (FLM) is still of great interest to the scientific community and the work of Gulina et al. presents an easy replicable method for the control of geometrical parameters of the induced tracks in lithium niobate.

Nevertheless, the authors do not discuss what technological impact these findings could provide for lithium niobate crystals in the numerous application they have mentioned in the introduction. The authors concentrate on the ‘observation’ of the outcome of a technique without offering much discussion on the desired geometry of the proposed structures and the possible improvements that could be made to exploit these findings in terms of their application on a scientific level.

Regardless, the work is interesting and the results may help further development of the described technique. I suggest publication only after addressing some minor issues that were spotted during the peer review process, and after the author provide future perspective for this technique, thus minor revision is recommended before publication in the Photonics journal by MDPI.

Specific comments

1.  As anticipated in the previous section, the author should provide a perspective on how these findings could be exploited in the various application fields they have mentioned.

2.  This is more of a suggestion than an issue, but it could be interesting to include in the introduction a small paragraph that justifies the use of the femtosecond pulsed laser for this particular experiment, as it provide a tool to perform the cold-processing for a vast class of materials, such as lithium niobate. The extremely reduced thermal effects during ultrashort pulses irradiation are very important for this type of processing.

3.  The laser repetition rate is missing in the abstract.

4.  Have the authors also changed the scanning velocity as a parameter? Could it also affect the geometrical properties of the tracks?

5.  Were figure 3c and b collected on the “cross-section”? If so it should be reported in the caption for clarity.

6.  Have the authors calculated a threshold pulse energy for which the tracks forms?

The manuscript is written in a clear way, using a good English. I did not find
